# GenPup-M: A novel validated owner-reported clinical metrology instrument for detecting early mobility changes in dogs

Natasha L. Clark[1]*, Karl T. Bates[1,2], Lauren K. Harris[3], Andrew W. Tomlinson[4], Jane K. Murray[4], Eithne J. Comerford[1,2,3]

**1** Department of Musculoskeletal & Ageing Science, University of Liverpool, Liverpool, United Kingdom, **2** Medical Research Council (MRC) and Versus Arthritis as part of the Medical Research Council Versus Arthritis Centre for Integrated Research into Musculoskeletal Ageing (CIMA), Northumberland, United Kingdom, **3** Dogs Trust, London, United Kingdom, **4** Department of Small Animal Clinical Sciences, Small Animal Teaching Hospital, Neston, United Kingdom

* Natasha.Clark@liverpool.ac.uk

## Abstract

### Objective

To use a previously validated veterinary clinical examination sheet, Liverpool Osteoarthritis in Dogs (LOAD) questionnaire, combined with kinetic and kinematic gait analysis in dogs with/without mobility problems to demonstrate the capacity of a novel clinical metrology instrument ("GenPup-M") to detect canine mobility impairments.

### Design

Quantitative study.

### Animals

62 dogs (31 with mobility impairments and 31 without mobility impairments).

### Procedure

The dogs' clinical history was obtained from owners and all dogs underwent a validated orthopaedic clinical examination. Mobility impairments were diagnosed in the mobility impaired group based on clinical history and orthopaedic examination. Owners were asked to complete GenPup-M along with a previously validated mobility questionnaire (Liverpool Osteoarthritis in Dogs (LOAD)) to identify construct validity. As a test of criterion validity, the correlation between instrument scores and the overall clinical examination scores, along with force-platform obtained peak vertical forces (PVF) were calculated. GenPup-M underwent internal consistency and factor analysis. Spatiotemporal parameters were calculated for dogs with/without mobility impairments to define the gait differences between these two groups.

**Data Availability Statement:** All relevant data are within the paper and its Supporting Information files.

**Funding:** This study was funded by a Canine Welfare Grant from Dogs Trust, UK. There was also funding through Centre for Integrated Research into Musculoskeletal Ageing (CIMA) for the motion capture cameras which came from a large equiptment grant from Medical Research Council, UK. The funders had no role in study design, data collection and analysis, decision to publish, or preparation of the manuscript.

**Competing interests:** The authors have declared that no competing interests exist.

## Results

Principal Component Analysis identified GenPup-M had two components with Eigenvalues >1 ("stiffness/ease of movement" and "willingness to be active/exercise"). Cronbach's α was used to test internal consistency of GenPup-M and was found to be "good" (0.87). There was a strong, positive correlation between GenPup-M and LOAD responses ($r^2 = 0.69$, p<0.001) highlighting construct validity. Criterion validity was also shown when comparing GenPup-M to clinical examination scores ($r^2 = 0.74$, p<0.001) and PVF ($r^2 = 0.43$, p<0.001). Quantitative canine gait analysis showed that there were statistically significant differences between peak vertical forces (PVF) of mobility impaired and non-mobility impaired dogs (p<0.05). Analyses of PVF showed that non-mobility impaired dogs more evenly distributed their weight across all thoracic and pelvic limbs when compared to mobility impaired dogs. There were also consistent findings that mobility impaired dogs moved slower than non-mobility impaired dogs.

## Conclusion and clinical relevance

GenPup-M is a clinical metrology instrument (CMI) that can be completed by dog owners to detect all mobility impairments, including those that are early in onset, indicating the versatility of GenPup-M to assess dogs with and without mobility impairments. Results of the study found that GenPup-M positively correlated with all three objective measures of canine mobility and consequently showed criterion and construct validity. Owner-reported CMIs such as GenPup-M allow non-invasive scoring systems which veterinary surgeons and owners can use to allow communication and longitudinal assessment of a dog's mobility. It is anticipated that GenPup-M will be used by owners at yearly vaccinations/health checks, allowing identification of any subtle mobility changes, and enabling early intervention.

## Introduction

Many factors can lead to a dog developing a mobility impairment [1–3]. Dogs with chronic osteoarthritis (OA) may be considered to have mobility impairments due to abnormal stresses causing chronic inflammation [4]. In veterinary medicine, when determining if an animal has a mobility impairment, it is important to assess the animal's ability to engage in daily activities and interactions, and to move or exercise freely. If an animal is unwilling to play, exercise or express normal behaviour without having pain or reduced movement, then it can be considered to have a mobility impairment [5]. Studies have also investigated the impact of impaired mobility on quality of life (QoL) in companion animals and found that owners perceive improvements in mobility with an increased QoL [6–8].

It is estimated the prevalence of conditions leading to mobility impairments poses a risk to canine long-term health and welfare [9]. A recent study calculated annual period prevalence at 2.5%, equating to 200,000 dogs in the UK affected by OA, which can cause subsequent mobility impairments [1]. Data collected from 200 veterinary surgeons in North America found that 20% of all dogs over one year old were suffering from an orthopaedic condition which caused mobility impairments, this subsequently increased to 80% by the age of eight years old [10]. Similarly, it has been estimated that up to 50% of UK dogs are diagnosed with some degree of reduced mobility between the ages of eight to thirteen [11]. A longitudinal study on 48

Labrador retrievers in 2006, showed that 15% of Labrador retrievers had radiographic evidence of hip OA by two years old, rising to 26% by five years old. Furthermore, 67% of Labrador retrievers were reported to suffer from OA by the end of their life [12].

Since the 1970s, diagnosis of orthopaedic disease has predominantly been based on visible radiographic changes, however, radiography has poor sensitivity for early pathological changes within the joint. Furthermore, it fails to achieve a correlation between identifiable musculoskeletal diseases (such as OA) and clinical joint function [13]. Hence, there is a need for a reliable, valid and reproducible measure to determine early onset OA [14]. Newer imaging modalities such as magnetic resonance imaging (MRI) and computed tomography (CT) are trying to reduce disparities between the early onset of pathology and initial OA diagnosis, since composition and structural cartilage changes can be detected (via MRI) alongside potential microscopic bone remodelling and osteophytosis (via CT) [15]. Similarly, high-field MRI imaging is being used to determine the biomarkers involved in the early OA disease process within a joint [15, 16]. Whilst radiography and advanced imaging are commonly used in the detection and diagnosis of OA, their use is not without risk, as dogs require a general anaesthetic to reduce movement artefact.

Additional non-invasive scoring mechanisms are available for veterinary surgeons to use in conjunction with imaging methodologies and clinical examination to allow longitudinal assessment of mobility [14]; these can be defined as subjective or objective [17]. Subjective scoring systems provide a method for owners and veterinary surgeons to communicate and measure limb abnormalities [18, 19]. Objective analysis can either be via kinetic and/or (two-dimensional or three-dimensional) kinematic gait evaluation [20–24]. Kinetic gait analysis typically involves the use of force plates to produce computational measurements of ground reaction force (GRF) [25, 26], peak vertical force (PVF) and vertical impulse (VI). Generally, dogs who have mobility impairments resulting in lameness have lower PVF and VI in the affected limb [27, 28]. Kinematic gait analysis requires specialist equipment and retro-reflective skin markers that are placed at specific anatomical landmarks [29]. In recent years, kinematic data collection has been used alongside kinetic gait analysis to provide more information about prognostic factors pre- and post-surgical intervention [30]. Similarly, gait analysis alongside clinical examination [31, 32], radiography [31, 33] or owner-reported questionnaires [21, 22] has allowed the detection and monitoring of early changes in canine mobility, enabled adequate analgesia, improved muscle strength and preservation of joint function [20, 33–35]. Clinical Metrology Instruments (CMIs) are subjective assessments of mobility. CMIs are questionnaires aimed to evaluate and address clinically relevant questions about a specific construct [36]. Orthopaedic CMIs used in the human medical field include: Knee injury and Osteoarthritis Outcome Score (KOOS) and Western Ontario and McMaster Universities Osteoarthritis Index (WOMAC); subsequently prompting CMI use in the veterinary profession [21, 37]. Veterinary CMIs for assessing canine OA have been validated, most commonly by comparison to similar CMIs, advanced imaging, kinematic or kinetic gait analysis [20, 22, 38] to provide information relating to reproducibility, accuracy and reliability [39, 40]. There are six widely used CMIs to determine chronic, longitudinal assessment of OA in dogs [20], including Liverpool Osteoarthritis in Dogs (LOAD), Helsinki Chronic Pain Index (HCPI), Canine Orthopaedic Index (COI),Canine Brief Pain Inventory (CBPI) and Canine Osteoarthritis Staging Tool (COAST).

Veterinary CMIs can prove useful when assessing the dog's quality of life as they can provide information on limb function and pain, something which radiography fails to achieve, since there are often disparities between clinical signs and radiographic severity [13, 41, 42]. Previous studies have also explored this disparity via veterinary- and owner-reported pain scores in comparison with radiographic findings using the HCPI [38]. Studies have also shown

that force plate analysis did not always correspond with radiographic severity [43, 44]. To date, there has been little work investigating the correlation between owner-reported CMIs and radiography in dogs. However, a recent study on working police dogs found that abnormal radiographic changes within the hip joint corresponded with worse CMI scores and reduced weight distribution [45]. Similarly, a cross-sectional study on 142 cats found that CMIs facilitate the diagnosis, evaluation and treatment of feline OA [35]. The results of these two studies suggest that CMIs can be used as a screening diagnostic test, with animals receiving poor CMI scores undergoing further diagnostic tests (e.g. radiography) to confirm the presence of OA.

Pre-validated CMIs are commonly used in veterinary practices [37] due to their open-access availability and accessibility for use in numerous musculoskeletal conditions. The current available CMIs only provide mobility assessments for dogs who have chronic orthopaedic conditions and are not designed for measuring subtle, early-onset changes in a dog's mobility. A 2023 study [46] investigated the use of LOAD and COI to determine differences between the pre- and post-operative stages in dogs (n = 125) with acute and chronic CCL disease/rupture. However, the first data collection point was 'pre-surgery', thus, there is not any data relating to any subtle changes within the dog's mobility before they became a surgical candidate, something which could be beneficial, especially in chronic CCL disease. The current study aims to validate a newly developed CMI ("GenPup-M") (S2 Fig). The GenPup-M mobility questionnaire was developed by two of the co-authors (E.C and J.M) with the intention of creating a novel CMI which could detect a range of canine mobility impairments, including early and subtle changes in mobility. GenPup-M is currently being used to collect mobility data from dog owners at repeated time points (from age 5 months) as part of the longitudinal Generation Pup™ study [47]. However, objective validation of GenPup-M has not yet been performed for use in either Generation Pup™ or the general dog population. Hence, the aim of this study was to validate the use of GenPup-M and provide evidence that GenPup-M can accurately identify changes relating to canine mobility reported by dog owners.

Newly developed questionnaires should ideally be validated for use in clinical settings [20]. The validity of a questionnaire is determined by analysing whether the questionnaire measures what it is intended to measure; validity is subdivided into four categories: Content, Face, Construct, and Criterion [48, 49]. Content validity refers to the extent in which the newly created CMI measures the appropriate content (e.g. questions relating to mobility), it also provides information about the variety of attributes that make up the desired construct, and if they are relevant and accurate [50]. Face validity determines if the items within the questionnaire measures the intended construct, this is usually achieved when experts review the instrument to ensure the construct (e.g. mobility impairments) is being tested [50, 51]. Construct validity investigates how well the newly developed instrument matches the true construct of what is being measured by comparing instruments against each other [20, 52]. Finally, criterion validity refers to the extent of which the newly designed instrument correlates with a previous standard, external measure [20, 50] and is usually compared to the gold standard method of analysis (e.g. quantitative gait analysis). Validity of an instrument also relies heavily on repeatability and consistency. Thus, repeatability is commonly examined via test-retest, during a time-frame where there is no change to the clinical condition of the underlying disease [5, 37, 53].

Our aim was to validate the GenPup-M questionnaire using similar methodology to previous veterinary mobility studies [20, 21, 37–39]. Prior to data collection, content validity was achieved by reviewing previous mobility CMIs (LOAD, CPPI and COI) to ensure GenPup-M contains questions to detect impaired mobility. Face validity was accomplished through veterinary epidemiologists and RCVS/ECVS specialists reviewing GenPup-M and ensuring the questions were logical and relevant to the study objectives. Throughout the study, criterion

validity was be achieved via objective gait analysis (PVF) [20] and a pre-validated veterinary clinical examination sheet [54]. Construct validity will be achieved by correlating GenPup-M responses to a previously validated canine mobility CMI, the Liverpool Osteoarthritis in Dogs (LOAD) which was selected due to its frequent use in veterinary mobility research [20, 55, 56].

Therefore, we hypothesised that there would be a significant correlation between GenPup-M and LOAD, and there would be a significant positive correlation between GenPup-M data and the objective measures of canine mobility impairments (veterinary clinical examination and peak vertical force (PVF)).

## Materials and methods

### Case selection

The study protocol was approved by the University of Liverpool Research Ethics Committee (VREC942). Due to the COVID-19 pandemic and the challenges of completing face-to-face research with members of the public, an online advertisement was run targeting staff members of the Institute of Life Course and Medical Sciences (ILCaMS) and the Small Animal Teaching Hospital (SATH), University of Liverpool (UK). Interested dog owners then contacted the principal investigators (N.C and E.C) and completed an online interview to assess the inclusion and exclusion criteria for enrolment in this study.

Following the online interview, dog owners were asked to sign a consent form and complete both CMIs (GenPup-M and LOAD) using an online internal questionnaire site (JISC.co.uk). For dogs with more than one caregiver, the same person was asked to complete both CMIs to reduce the effects of between-observer differences. Following questionnaire completion, owners were invited to the gait laboratory facilities based at the William Henry Duncan Building, University of Liverpool. During this time, the dog's clinical history was collected and an orthopaedic and neurological clinical examination was completed and recorded on the validated clinical examination sheet [54] (S1 Fig).

### Inclusion and exclusion criteria

The inclusion criteria allowed enrolment of all dogs over five months of age in the study if the consulting veterinary surgeons (N.C and E.C) and the owner had no concerns about the dog being subject to an independent veterinary assessment for mobility and gait analysis, based on health and/or behaviour concerns. The exclusion criteria were dogs which were receiving pain relief medication (for example non-steroidal anti-inflammatory drugs (NSAIDs)) for mobility or other health-related problems which prevented any mobility impairments, that would otherwise be expected to be evident on veterinary examination or gait analysis, from being masked. However, any dogs receiving nutraceuticals were included in the study as it has been previously suggested that this would not disguise any mobility impairments [20, 57]. Specific questions regarding if dogs were receiving any medication, which could potentially hide mobility impairments, were included in the GenPup-M questionnaire. Furthermore, no dogs who participated in this study were recently adopted, since GenPup-M relies on owner interpretation of mobility and this would be unknown if the dog was recently adopted.

### Questionnaire responses

Due to COVID-19 restrictions, dog owners were not permitted to be present during the clinical examination or gait analysis, therefore, completion of the GenPup-M and LOAD questionnaires had to be done remotely. Electronic versions of GenPup-M and LOAD were sent out to 52 owners via the Joint Information Systems Committee (JISC.co.uk) to their e-mail. The

online version of GenPup-M and LOAD were direct replicas to the original paper formats. Owners were asked to complete the LOAD questionnaire after GenPup-M to reduce respondent bias and ensure that the owners would read the questions carefully and not predict answer options [37]. Previous literature does not report any detrimental effects of completing questionnaires in a non-sequential manner [58–60]. The questionnaire data was exported from the JISC software into Excel spreadsheets to be analysed (Microsoft Excel 2016, Microsoft). All data corresponding to mobility function were pseudonymised. Dog names were also removed and replaced with participant identification numbers (PIDs) ranging from one to sixty-two.

The LOAD questionnaire contained an option to create a tabulated overall mobility score based on owner responses, ranging from normal to severe. However, GenPup-M did not contain a scoring system, therefore, one was created (based on scoring systems for the LOAD and CBPI) for this study to allow construct validity to be assessed. GenPup-M contains a total of twenty-four questions for owners to complete, including questions relating to supplements, terrain, frequency of walks, duration of walks and restriction of exercise (S2 Fig). However, only the 10 scale-rank questions relating to the dog's demeanour and tolerance to exercise were used to assess construct validity; the adopted scoring system can be found in S5 Fig. The highest attainable score for the GenPup-M questionnaire was 108 points: 0–27 = no concerns, 28–54 = mild concerns, 55–82 = moderation concerns, 82–108 = severe concerns for mobility. Mobility categories and total scores for GenPup-M were generated based owner responses from the ten Likert-style scales. The LOAD, CBPI, HCPI and COI also use numerical rating scales (NRS) and visual analogue scales (VAS) to create a score which increases with increasing degrees of disability and lameness [61–66].

## Clinical examination

The clinical examination sheet [54] (S1 Fig) contained a participant identification (PID) number corresponding to the individual dog; dog age, breed, sex and neuter status were also recorded. Bodyweight in kilograms (Kg) and body condition score (BCS) was entered using the 1–9 scale adapted by Laflamme [67] (S1 Fig).

The veterinary clinical examination was undertaken by N.C (with initial training by an ECVS boarded specialist (EC)) using the same methodology as Harris et al. [54]. Clinical history was obtained from owners and the researchers were blinded to the clinical examination results to prevent bias. The validated clinical examination sheet was a composite scoring system to standardise veterinary assessment of OA. All parts of the checklist could be categorised as follows: (a) mobility, an 11-point lameness score (0–10) adapted from Vasseur and Slatter [68], and scales rating the dog's ability to stand up and sit down (0–3); (b), number of joint abnormalities observed (e.g. crepitus, effusion); (c) JFS, joint function score (adapted from Impellizeri [69]); and (d) global score of severity of the dog's disease by the veterinary surgeon (none, mild, moderate, severe; 0–3). All dogs also underwent a brief neurological assessment; cranial nerves, menace and tracking reflexes were assessed during the head and neck component of the examination, and knuckling reflexes were assessed for all four limbs.

The maximum obtainable score for a severely mobility impaired dog using the above methodology was 16. Dogs were categorised as 0 = no concerns, 1–5 = mild concerns, 6–11 = moderate concerns and 12–16 = severe concerns for mobility [54].

## Gait analysis

**Kinetic and kinematic gait analysis.** Time-synchronised force plate recordings (Type 9260AA, Kistler UK) and 3D spatiotemporal kinematics were recorded for all dogs recruited

for this study. Dogs were allowed a minimum of five minutes to acclimatise to their surroundings and 5–10 "practise" trials were performed to enable the dog to understand the task. Dogs were trotted across the force plates approximately 25 times by the same handler to remove any interference [28, 34, 70–72]. Any trial where constant, steady-state locomotion in a straight line did not occur was deleted and repeated [28].

For the current study, peak vertical force (PVF) was the primary gait metric used to assess the criterion validity of the GenPup-M questionnaire. However, this data was also used in association with spatiotemporal parameters to assess individual canine gait 'within' the participant cohort and 'between' participants to establish the quantitative nature of gait differences between healthy and mobility impaired dogs. Twelve high-speed infra-red motion cameras (Qualisys Oqus-7) recorded spatiotemporal kinematic data via four retro-reflective markers. The motion capture cameras were ceiling mounted and arranged in an oval circumference with the three force plates in the centre. Before each trial, all Qualisys Oqus-7 cameras were calibrated using a calibration bar and wand.

Three retro-reflective markers were placed on the cervical vertebrae 7 (C7), thoracic vertebrae 13 (T13) and lumbar vertebrae 7 (L7). These three markers were used to calculate the average speed within in each trial. In addition, leg length measurements (in cm) for thoracic and pelvic limbs were taken for each dog for comparison against Froude number. Kinematic data was collected during the 25 trotting trials alongside kinetic data.

Dogs which had mobility impairments identified on clinical examination were classed as 'mobility impaired'. Dogs within this category were further divided into three more groups: (1) thoracic mobility impairments, (2) pelvic mobility impairments or (3) both thoracic and pelvic mobility impairments to allow comparisons between groups. Comparisons of kinetic and kinematic data were conducted across cohorts (i.e. non-mobility impaired versus mobility impaired dogs) after size-normalisation of raw/absolute data. Normalised data were obtained by the following calculations: speed was normalised into a Froude number [(Froude = speed x speed) / (gravity x leg length)] [73].

## Statistical analysis

**1) GenPup-M.** Ten scale/rank questions within the GenPup-M questionnaire (S2 Fig) were compared to a validated veterinary clinical examination sheet [54] (S1 Fig) Using the previous methodology, the GenPup-M questionnaire scores were also correlated with kinetic gait analysis (Peak Vertical Force (PVF)) to aid the assessment of criterion validity [20, 21, 37]. The ratio between thoracic and pelvic PVF was used to assess the correlation between GenPup-M responses and quantitative gait analysis. Construct validity was determined by comparing GenPup-M scores against the LOAD questionnaire.

Previous work used factor analysis to identify question weightings to determine if one or more of the main components predicted the absence/presence of mobility issues in the cross-sectional sample of dogs [20]. Hence, ten scale/rank questions within GenPup-M were used for factor analysis. A Kaiser-Mayer-Olin measuring of sampling adequacy ($>0.6$) was used. Eigenvalues of $>1$ [74], scree-plot analysis and theoretical interpretability enabled extracted factors to be assessed. Varimax-rotated models were used on extracted components to assess the item loading of factor analysis. Communalities had a cut-off value of 0.4 [20]. The untransformed variables were not normally distributed, therefore, Spearman's rank correlation coefficient ($p \le 0.05$, two-tailed) was used to explore the comparison between GenPup-M and the previously validated LOAD questionnaire scores, alongside GenPup-M and clinical examination scores and PVF. A Spearman's rank correlation test was used to assess the relationship between the 10 scale/rank GenPup-M responses and canine gait analysis parameters [75].

Based on previous literature [20], PVF was determined to be the most useful to assess this correlation. The significance level was set at p≤0.05 (two-tailed).

A Mann-Whitney U test was used to determine the sensitivity of GenPup-M by comparing mobility impaired versus non-mobility-impaired scores. Standard error of skewness, and subsequently, the degree of skewness was analysed to see if the data were significantly positively or negatively skewed by comparing the numerical value of "skewness" with twice the standard error of skewness for both negative and positive values. Finally, Cronbach's Alpha test was used to determine internal consistency within the cohort for GenPup-M responses [76].

**2) Clinical examination.** A Mann-Whitney *U* statistical test was used to determine if there was a statistical difference in overall clinical examination scores between mobility impaired and non-mobility impaired dogs. A Spearman's rank correlation test was used to assess the relationship between age and severity of mobility. A chi-square test of independence was performed on body condition score (BCS) to examine the relationship between BCS and overall clinical examination score.

**3) Kinetic and kinematic gait analysis.** Statistical comparisons for force plate and spatiotemporal data were undertaken for mobility impaired and non-mobility impaired dogs. Significance levels were set to an alpha of p≤0.05. ANOVAs were used for spatiotemporal data analysis as data were normally distributed (confirmed by Shapiro-Wilk tests). To investigate the possible interaction between these changes in spatiotemporal gait changes (particularly speed and stride length) and body size in the two cohorts, we used ANCOVAs of Froude numbers and stride lengths against body mass and average limb lengths (as overall body size) to compare slopes (Figs 4 and 5).

## Results

### Dog demographics

Seventy-one dogs were recruited for the study, however, three did not meet the inclusion criteria due to being placed on analgesics before data collection. A further six dogs (four Chihuahuas and two Shih-Tzus) were excluded from the analysis due to having an incomplete data set. Hence, the total number of dogs that were included in the data analysis for this study was 62 (n = 31 mobility impaired and 31 non-mobility impaired). Demographic data of the cohort are summarised in Table 1.

### Veterinary clinical examination

Of the n = 62 dogs that were assessed by veterinary clinical examination, 31 were documented to have a mobility impairment. Of the 31 with mobility impairments recorded, only 13 of these dogs' owners reported a history or definitive diagnosis of mobility impairment (confirmed either by veterinary clinical history or previous radiography). The highest overall combined clinical examination score could be sixteen (excluding BCS). Table 2 provides an overview of the overall clinical examination score that occurred for mobility impaired and non-mobility impaired dogs. No dogs were found to have neurological deficits resulting in a mobility impairment or reduced quality of life.

A Mann-Whitney U statistical test was used to determine if there was a statistical difference in overall clinical examination scores between mobility impaired and non-mobility impaired dogs. Mobility impaired dogs had significantly higher clinical examination scores than non-mobility impaired dogs (U value = 496.0, Z = -7.252, p <0.001).

The impact of age, BCS, and breed on the degree of mobility impairment were investigated by using data obtained from the validated clinical examination sheet. Age had a significant positive correlation with an increase in overall clinical examination scores ($r^2$ = 0.66, P<0.001)

**Table 1. Demographic data of the mobility impaired (n = 31) and non-mobility impaired dogs (n = 31) cohort.**

| Variable | | Mobility Impaired | Non-Mobility Impaired |
|---|---|---|---|
| Gender (number) | Male (%) | 2 (6.5%) | 2 (6.5%) |
| | Neutered Male (%) | 15 (48.4%) | 7 (22.5%) |
| | Female (%) | 0 (0%) | 8 (25.8%) |
| | Neutered Female (%) | 15 (48.4%) | 13 (41.9%) |
| Age (Months) | Mean (SEM) | 97 (6.91) | 47 (5.01) |
| | Minimum | 17 | 5 |
| | Maximum | 192 | 105 |
| Bodyweight (Kg) | Mean (SEM) | 20.5 (1.81) | 22.2 (1.75) |
| | Minimum | 6.4 | 7.1 |
| | Maximum | 37.2 | 36.2 |
| Body Condition Score (BCS) | Mean | 4 | 4 |
| | Minimum | 4 | 3 |
| | Maximum | 7 | 6 |
| Breed (number) | Number of breeds represented (no, %) | 15 | 12 |
| | Cross Breed (%) | 6 (19.4%) | 7 (22.6%) |
| | Labrador Retriever (%) | 5 (16.2%) | 7 (22.6%) |
| | Small Terrier breeds (%) | 5 (16.2%) | 3 (9.7%) |
| | English Springer Spaniel (%) | 3 (9.7%) | 3 (9.7%) |
| | Border Collie (%) | 2 (6.5%) | 1 (3.2%) |
| | Other (%) | 10 (32.3%) | 10 (32.3%) |
| Orthopaedic examination (number) | Number of joints affected (no, %) | 31 | 0 |
| | Elbow (%) | 8 (25.8%) | |
| | Stifle (%) | 4 (12.9%) | |
| | Hip (%) | 8 (25.8%) | |
| | Carpus (%) | 4 (12.9%) | |
| | Tarsus (%) | 0 (0%) | |
| | Shoulder (%) | 0 (0%) | |
| | Combined thoracic and pelvic limb impairments (%) | 7 (22.6%) | |

Gender, age, body weight, body condition score, breed and joint affected is shown. Percentages for each category is also represented for the data set. The joint affected is included for the mobility and non-mobility impaired cohorts.

**Table 2. This table summarises of the frequency at which the overall clinical examination scores occurred for dogs with and without mobility impairments (excluding the BCS score values).**

| Mobility Impaired | Mobility Category | Overall Clinical Examination Score | Frequency (n =) | Percentage (%) | Cumulative Percentage |
|---|---|---|---|---|---|
| **No** | No concerns | 0 | 31 | 100.0 | 100.0 |
| **Yes** | Mild concerns | 1–5 | 24 | 77.4 | 77.4 |
| | Moderate concerns | 6–11 | 6 | 19.4 | 96.8 |
| | Severe concerns | 12–16 | 1 | 3.2 | 100 |
| | | Total | 31 | 100.0 | 100.0 |

The lowest score of four mobility impaired dogs was 3/16, and the highest score given for one dog was 14/16. All non-mobility impaired dogs had a clinical examination score of 0. The clinical examination scoring system was used to categorise mobility impaired dogs and non-mobility impaired dogs, alongside the severity of the mobility impairments [54].

and a significant positive correlation (p<0.01, two-tailed) with stand-up/sit-down and joint function score (JFS) (0.63, 0.58, 0.61, and 0.70, respectively).

A chi-square test of independence was performed on body condition score (BCS) to examine the relationship between BCS and overall clinical examination score. The relationship between these variables was not significant, $X^2$ (36, N = 62) = 23.652, p = 0.943. Thus, BCS did not affect the overall degree of mobility impairment in this study.

## GenPup-M validation

**Construct validity.** *GenPup-M and LOAD responses.* Factor analysis on the GenPup-M questionnaire revealed two components with Eigenvalues >1 and thus, the ability of GenPup-M to detect two aspects of a dog's mobility; "stiffness/ease of movement" and "willingness to be active/exercise".

The GenPup-M and LOAD questionnaire responses were correlated against one another using Spearman's rank correlation. There was a significant, positive relationship between total GenPup-M and LOAD scores. Spearman's correlation coefficient between GenPup-M and LOAD was $r^2$ = 0.68 (p<0.001) showing there was a strong significant correlation for the inter-instrument comparisons (> 0.68–1 = a strong correlation [77]). A scatterplot of GenPup-M versus LOAD is shown below (Fig 1). Kendall's tau_b was also used to investigate concordance and discordance more conservatively, as Kendall's tau is generally smaller when detecting a gross error of sensitivity (Kendall's tau_b = 0.528 (p<0.001)).

**Criterion validity.** *GenPup-M responses and veterinary clinical examination scores.* A Spearman's rank correlation coefficient was used to determine if the clinical examination scores (obtained from the sum of scores on the 11-point lameness score (0–10), stand-up and sit-down (0–3) and joint function score (0–3)) and GenPup-M responses correlated with each

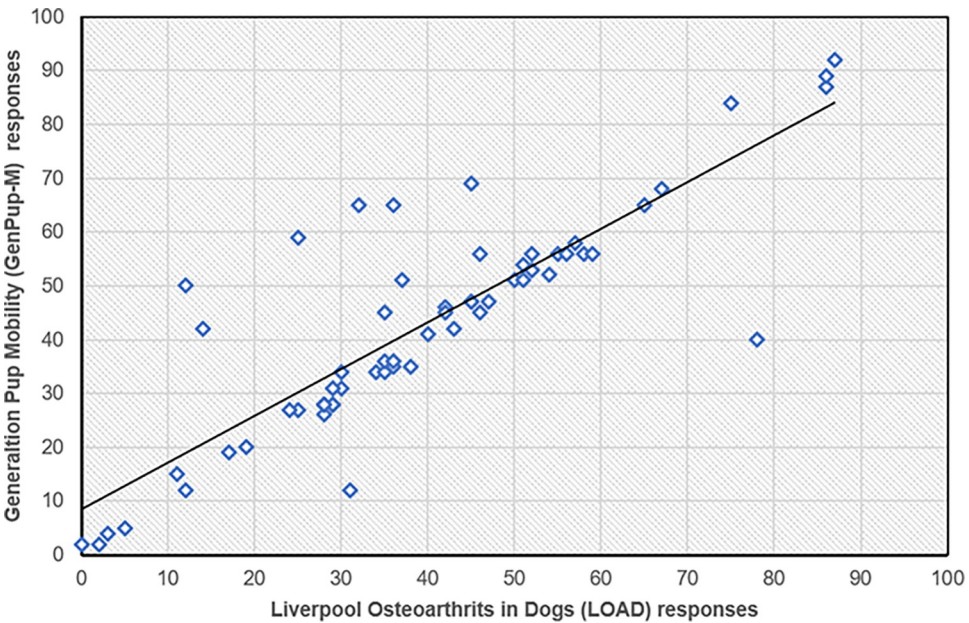

**Fig 1. GenPup-M responses versus Liverpool Osteoarthritis in Dogs (LOAD) questionnaire scores used to determine construct validity.** There was a significant, moderate positive correlation typical of inter-instrument comparisons showing that GenPup-M can accurately measure mobility impairments in dogs when compared to a pre-validated mobility questionnaire (LOAD). Spearman's Rank Correlation Coefficient = 0.69 (p<0.001). A black line of best fit is shown through the calculated overall scores obtained from GenPup-M and LOAD questionnaires. GenPup-M and LOAD responses can be found on the X- and Y-axis, respectively.

other. There was a significant positive correlation between total GenPup-M questionnaire scores and clinical examination scores. Spearman's correlation coefficient between GenPup-M and veterinary clinical examination was $r^2 = 0.74$ (p<0.001) showing there was a significantly strong, positive correlation between this data (Fig 2). Furthermore, higher GenPup-M scores were associated with higher clinical examination scores (p<0.05) indicating that there was an agreement between veterinary and owner assessment of impaired mobility.

*GenPup-M responses and Peak Vertical Forces (PVF).* There was a moderate, positive correlation between GenPup-M responses and PVF ($r^2 = 0.43$, p<0.001) [70]. Kendall's tau-b also showed a moderate positive correlation of 0.49 (p<0.001) between GenPup-M responses and PVF (Fig 3). There was also a strong positive correlation between PVF and the overall veterinary clinical examination scores ($r^2 = 0.51$, p<0.001). Kendall's tau-b found the agreement between PVF and veterinary clinical examination to have a moderate positive correlation of 0.50 (p<0.001).

**Gait data.** *Kinetic (peak vertical force (PVF)) analysis.* In the following sections, the data are expressed as mean (standard error mean) and ± standard deviation. Non-mobility impaired dogs generally had higher normalised thoracic and pelvic PVF values than mobility impaired dogs (Fig 4A). Both these differences were found to be statistically significant. Dogs with pelvic mobility impairments had the lowest pelvic PVF (0.65 (0.04) + 0.16) compared to those with thoracic mobility impairments (0.69 (0.02) ± 0.13) or combined thoracic and pelvic mobility impairments (0.83 (0.04) ± 0.12) (Fig 4B). Non-mobility impaired and mobility impaired dogs were recovered as statistically significantly different to each other in their thoracic to pelvic limb PVF ratio, with a higher ratio (i.e. relatively higher thoracic to pelvic PVF) in non-mobility impaired dogs (Fig 4).

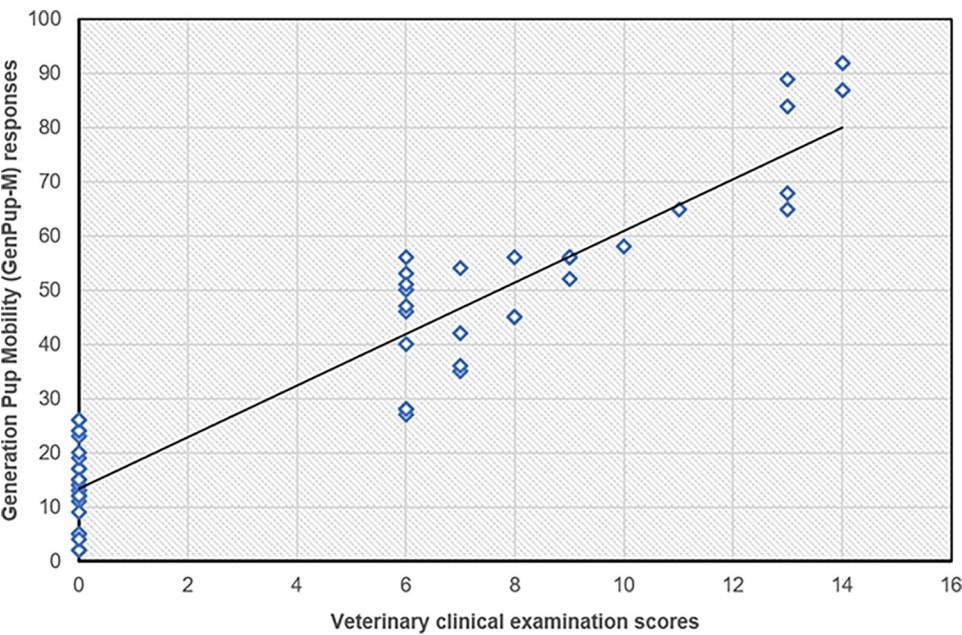

**Fig 2. GenPup-M versus the pre-validated clinical examination sheet [50] used to standardise the veterinary assessment of OA.** Scores were obtained from the following: (a) mobility, an 11-point lameness score (0–10) [64], and scales rating the dog's ability to stand up and sit down (0–3); (b) JA, number of joint abnormalities observed (e.g. crepitus, effusion); (c) joint function score (JFS) [65]; and (d) global score (GS) of the severity of the dog's disease by the veterinary surgeon (none, mild, moderate, severe; 0–3). There was a significant, moderate positive correlation between the GenPup-M owners' responses and veterinary clinical examination scores highlighting good criterion validity. Spearman's Rank Correlation Coefficient = 0.74 (p<0.001). A black line of best fit is shown through the scatterplot. Clinical examination scores and GenPup-M responses can be found on the X- and Y-axis, respectively.

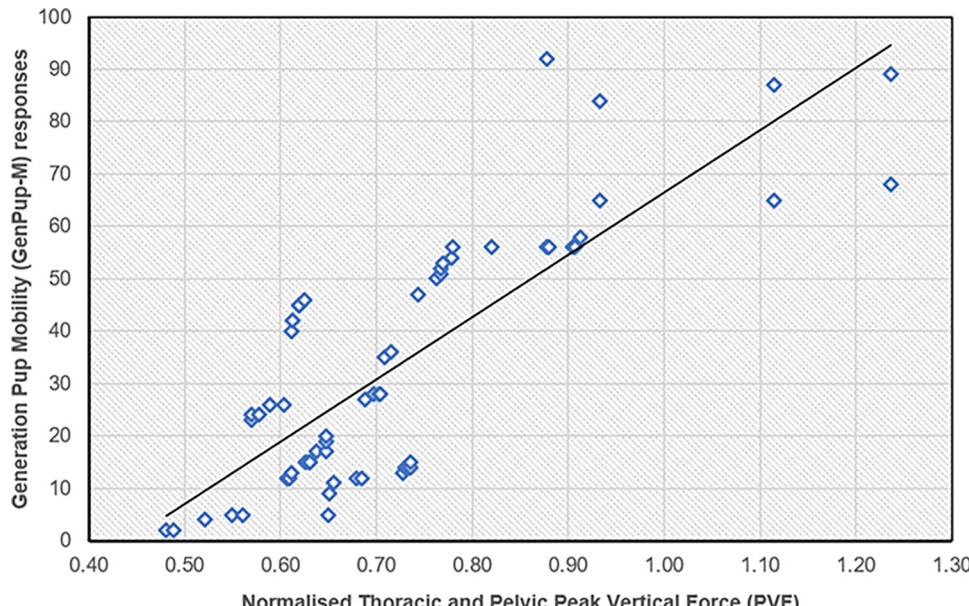

**Fig 3. GenPup-M questionnaire responses versus normalised thoracic and pelvic peak vertical forces (PVF) used to determine criterion validity.** PVF were obtained from kinetic gait analysis on all 62 dogs enrolled on the study and were normalised to body weight (Kg). There was a moderate positive correlation highlighting good criterion validity for GenPup-M questionnaire. Spearman's Rank Correlation Coefficient = 0.43 (p<0.001). A line of best fit is also shown on the scatter plot. GenPup-M and PVF values can be found on the X- and Y-axis, respectively.

*Kinematic analysis.* The average speed for each dog across their 25 trials was combined to calculate an average speed for non-mobility and mobility impaired groups. Speed values were obtained from by averaging values from three retroreflective markers (on C7, T13, L7) and normalised using the calculation [(Froude = speed x speed) / (gravity x leg length)] [73]. Dogs in the non-mobility impaired category (mean absolute speed 1.74m/s; mean Froude 0.80

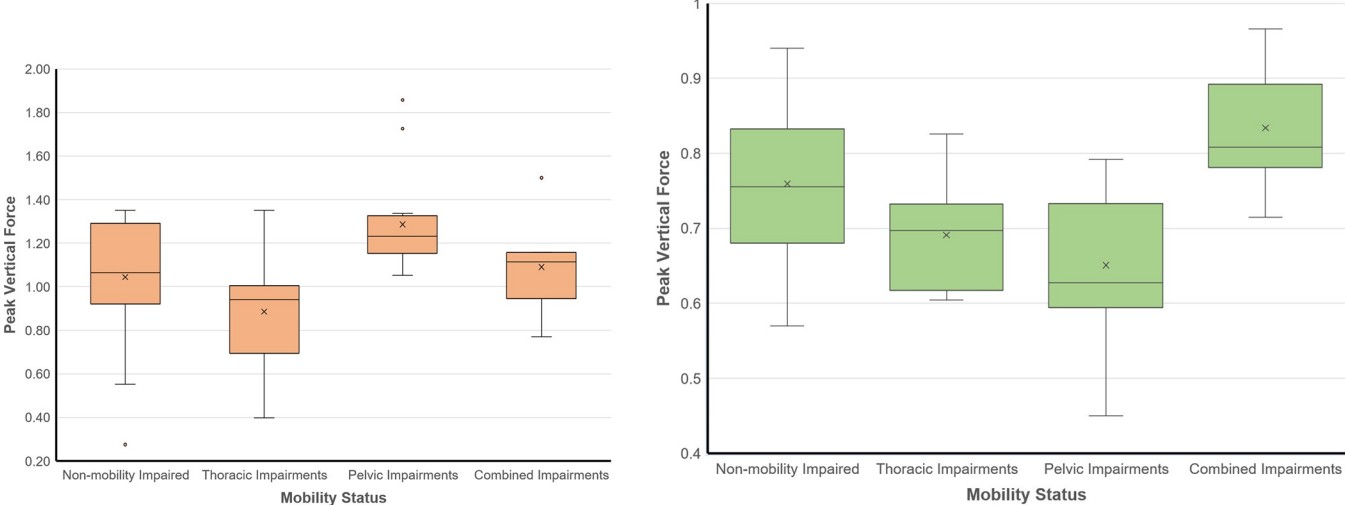

**Fig 4.** Peak vertical force (PVF) was normalised using body weight (A) thoracic and (B) pelvic limbs across all mobility and non-mobility impairment categories. Each category is represented by PVF (normalised by body mass). 'X' represents the mean, the whiskers (vertical lines) extend from the ends of the box to show the minimum value and maximum values (excluding outliers), black lines represent the median of the whole dataset values and separates the 1st and 3rd quartile. Outliers are highlighted by solid dots.

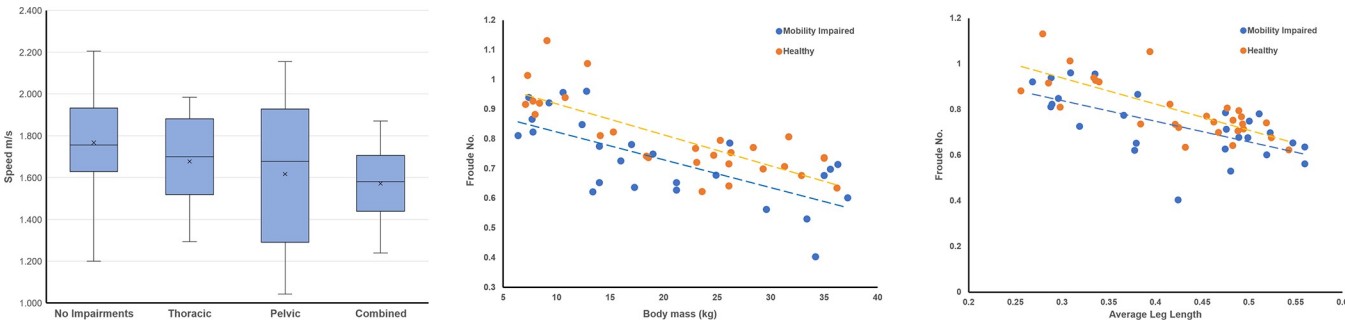

**Fig 5. Comparisons of absolute and normalised speeds and stride lengths in non-mobility and mobility impaired dogs.** On average, non-mobility impaired dogs moved at higher (A) absolute and (B-C) size-normalised speeds. This difference was recovered, with similar magnitude, when examined relative to differences in (B) body mass and (C) average leg length.

**Table 3. ANCOVA results to show the possible interaction between normalised speed differences.**

| Variable | ANCOVA result |
|---|---|
| Body Mass | F (1,50) = 64.509, p = 0.039 |
| Average Leg Length | F (1,50) = 1.704, p = 0.019 |

Body size, using both body mass and average limb lengths are measures of overall body size within this group.

(0.03) ± 0.28) walked faster in both absolute and normalised terms than all mobility impaired categories (mean absolute speed 1.62m/s (0.1) ± 0.22; mean Froude 0.73 (0.09) ± 0.13) (Fig 5). Dog with thoracic-only impairments (mean absolute speed 1.68m/s (0.08) ± 0.21; mean Froude 0.73 (0.04) ± 0.14) walked faster than dogs with combined thoracic and pelvic limb impairments (mean absolute speed 1.56m/s (0.09) ± 0.24; Froude 0.71 (0.01) ± 0.06) and pelvic-only limb impairments (mean absolute speed 1.61m/s (0.08) ± 0.33; Froude 0.73 (0.03) ± 0.16) (Fig 5).

An ANOVA test confirmed that absolute speeds were significantly higher in non-mobility impaired cohort than in mobility impaired dogs (F (1,55) = 4.938, p = 0.027). ANCOVAs were used to investigate possible interaction between these normalised speed differences and body size, using both body mass (Fig 5B) and average limb lengths (Fig 5C) as measures of overall body size (Fig 5, Table 3). These revealed that mobility impaired dogs moved at significantly lower Froude numbers across all body sizes (Fig 5B and 5C), with slopes that are homogeneous and met the assumption of homogeneity of regression. Levene's Test of Equality of Error Variances showed the variances were homogenous across both groups. However, there is some (qualitative) suggestion of a relatively greater difference in Froude numbers in dogs with shorter limbs (Fig 5C).

## Discussion

This study aimed to validate a novel owner-reported mobility questionnaire ("GenPup-M") designed to detect early onset changes in mobility. GenPup-M was validated by using a veterinary clinical examination sheet [54] and quantitative canine gait analysis for criterion validity, alongside a pre-validated owner-reported mobility questionnaire (LOAD) for construct validity. Data from the current study found that GenPup-M can detect changes in a dog's mobility and contribute towards the early detection and management of canine musculoskeletal disease.

The validity of a questionnaire or clinical metrology instrument is important as it determines if the questionnaire measures what it is intended to by comparing the responses against similar constructs or a gold-standard method. For example, advanced imaging (CT, MRI), and kinematic or kinetic gait analysis [21, 22, 38] to provide accuracy relating to reproducibility, accuracy and reliability of the questionnaire [39, 40]. Factor analysis on the GenPup-M questionnaire revealed two components with Eigenvalues >1 and thus, the ability of GenPup-M to detect two aspects of a dog's mobility; "stiffness/ease of movement" and "willingness to be active/exercise". These components are important for the overarching objective of GenPup-M to detect early onset mobility impairments as an animal's eagerness to be active or exercise may be reduced if there is a subtle mobility impairment present [17, 21, 22, 37, 78]. In contrast, the LOAD questionnaire retrieved three components; (1) "stiffness/mobility impairment", (2) "activity/exercise" and (3) "attitude/demeanour". Previous work has shown that component three of the LOAD has identified the "effect of weather" and not "attitude/demeanour" [20]. This difference may be accounted for by a change in participants, since most dogs were only reported to have a mild lameness (n = 24), and that the research was completed during spring and summer months, hence, the effect of cold, damp weather was not at the forefront of owner's interpretations. Nonetheless, these components will be important in the longitudinal monitoring of chronic OA in dogs, which the LOAD questionnaire was designed and developed to be used for [20]. This may also explain why, in previous studies, the Helsinki Chronic Pain Index (HCPI) and Canine Brief Pain Inventory (CBPI) had a stronger positive correlation with the LOAD questionnaire ($r^2$ = 0.74 and $r^2$ = 0.76, respectively (Walton et al., 2013)) compared to the GenPup-M ($r^2$ = 0.68), since HCPI, CBPI and LOAD are all designed to provide monitoring for chronic OA.

A validated veterinary clinical examination sheet [54] was used to categorise mobility impairments for this dog cohort into either mobility impaired or non-mobility impaired. Our study used a similar methodology to previous research [20, 37, 38] where clinical history and examination data were obtained to include/exclude participants in the study. To mitigate error, several factors were taken into consideration when assigning the groups: pain elicited when touching a joint, tissue swelling, muscle atrophy and crepitus. Whereas, these factors may have been overlooked in previous work when discussing the limitations of diagnosing mobility impairments via visual gait analysis [19, 79]. Moreover, dogs who have unilateral gait asymmetries often only have one aspect of impaired limb function, whereas, dogs with bilateral disease can have symmetrical abnormal load-bearing due to compensatory mechanisms [38, 79]. Hence, dogs with more severe mobility impairments may have good mobility but marked asymmetry.

The relatively strong correlations between gait metrics and questionnaire responses suggest that owners can reliably recognise signs of mobility issues in their dog's habitual movement patterns; only 5 out of the 13 dogs had moderate or severe mobility impairments (n = 5 and n = 1, respectively), with 8 dogs having mild mobility impairments. Furthermore, 8 out of the 31 dogs presented with veterinary clinical history suggestive of mobility impairment, with a further 6 dogs having diagnostic radiographic evidence of joint OA or stifle joint pathology relating with CCL disease. Additionally, this highlights the ability of owners to detect subtle abnormalities within their dogs' gait and subsequently present them to veterinary practices for further evaluation. The overarching aim of GenPup-M is for it to be used to detect changes in mobility through use at repeated timepoints throughout the life of a dog, (starting when dogs are 5 months old, when most dogs have no mobility impairments). Thus, GenPup-M will be the first veterinary mobility CMI that is relevant to dogs with and without mobility problems. As newer research is investigating the positive correlation between worse CMI scores and radiographic pathology [37, 45], GenPup-M could be used as screening diagnostic tool, with

emphasis on dogs retrieving poor scores undergoing further diagnostic tests (e.g. radiography, CT or MRI).

In this study, mobility impaired dogs walked slower in absolute and relative terms than non-mobility impaired dogs, and this was achieved primarily by taking shorter strides (Figs 4 and 5). The lowest speeds and stride lengths were recorded in dogs with combined limb mobility issues (i.e. dysfunction in both thoracic and pelvic limbs), and we also find clear evidence for a differential gait response in other parameters according to the location of the mobility impairment. All non-mobility impaired had statistically higher thoracic than pelvic limb PVFs, which is consistent with most previous studies [80–85]. In dogs with only thoracic limb impairments, there was some tendency to redistribute load away from the affected limb. However, the magnitude of load redistribution away from the impaired limb was considerably greater in dogs with only pelvic limb impairments (Fig 4), as these dogs also had the greatest limb asymmetry of any group. This shift in load distribution appears to be associated with more equal cycle and swing times between the limbs, a relatively low thoracic limb duty factor and stance time, and a slightly higher thoracic stride length relative to the pelvic limb in these dogs (Fig 5).

Dogs placed in the pelvic category mainly had some degree of hip dysplasia, previous research has shown that hip dysplasia can redistribute loading away from the affected limb [31, 33, 86]. As discussed above, the greatest load redistribution was in the pelvic mobility impaired cohort, and subsequently affected other gait parameters. Unilateral elbow pain and previous carpal fractures were among the most common orthopaedic conditions to represent the thoracic impaired group. However, dogs with previous carpal fractures had the fastest absolute and relative speeds, suggesting that previous injuries to the carpal joint did not affect the dog's mobility once fully healed. Although sub-dividing dogs based on their affected limb leads to relatively small sample sizes, the nature of differences recovered here (e.g. a suggestion of redistribution of the load away from the dysfunctional limb) suggests that such categorisation or data segregation would be beneficial in future studies.

During this study, GenPup-M was tested for criterion and construct validity, and was found to positively correlate with a validated clinical examination sheet ($r^2 = 0.74$) [54], a previously validated mobility CMI ($r^2 = 0.69$) [22] and quantitative gait analysis ($r^2 = 0.43$). GenPup-M was also found to have good internal consistency (Cronbach's $\alpha = 0.87$) and the ability to detect three components of reduced mobility in dogs. The strong correlation with the clinical examination was considered a positive finding, as it is our aim that GenPup-M will be mostly used alongside clinical history and clinical examination in small animal practice. Interestingly, GenPup-M correlated more positively with PVF, compared to when LOAD responses were correlated with PVF ($r^2 = 0.23$) [22]. Perhaps, this is due to the GenPup-M questionnaire being more sensitive to detecting more mild and subtle changes in canine gait (which formed the majority of the dogs in this cohort (n = 24)). Of course, except for 12 dogs who presented with veterinary clinical history (n = 8) and radiographic evidence of musculoskeletal disease (n = 6), there was no definitive diagnosis for the dogs in the mobility impaired cohort, only presumptive. However, it is important to note that mild, moderate and severe mobility impairments were detected on clinical examination, gait analysis and GenPup-M. Furthermore, GenPup-M is not acting as a diagnostic tool, its primary aim is to identify mobility impairment both within research studies such as the Generation Pup™ study, and for owners who can then highlight concerns with their dog's mobility with their veterinary surgeon. Hence, it was not essential that the mobility impaired cohort had definitive diagnoses as many dogs will undergo further diagnostic imaging before a diagnosis is made.

## Limitations

Waxman et al. [19], stated that force plate analysis was the gold standard approach for quantifying mobility problems. However, there were some limitations of gait analysis which arose during this study, particularly when relating the gait data to clinical examination data (which was used to categorise dogs as mobility impaired or non-mobility impaired). Furthermore, for the current study, joint kinematics were not investigated due to the restricted time frame of this study. Another limitation is perhaps inter-breed variation; the cohort in this study consisted of 17 different breeds ranging from Jack Russell Terriers to a Bullmastiff. Therefore, we lacked the overall sample size to use statistical approaches (e.g. mixed effect models) that can provide information on the effect of confounding variables on our dataset. Other gait studies have focused on more homogenous groups such as Labrador retrievers [12, 21, 79, 87] or Greyhounds [88–90]. There was a consensus among these studies that more variance was attributable to individuals when comparing different breeds and identical breeds [29, 70, 91]. Thus, morphology and conformation can be highly influential in interpreting results from canine gait analysis. However, in this study, there was no predominant breed in either the mobility impaired or non-mobility impaired category.

Radiography is the still mainstay diagnostic choice for orthopaedic conditions, mainly due to its ease of use and availability [12, 92, 93]. However, previous work has suggested that radiography has poor sensitivity for early pathological orthopaedic changes and fails to achieve a correlation between radiographic signs of joint dysfunction and clinical presentation [13]. Radiography was not undertaken during the study due to animals requiring an invasive general anaesthetic to obtain diagnostic images. Furthermore, a validated veterinary clinical examination sheet [54], combined with kinetic and kinematic gait analysis was deemed sensitive enough to determine if a dog had a mobility impairment [19, 20, 37]. Newer imaging modalities such as magnetic resonance imaging (MRI) and computed tomography (CT) are trying to reduce disparities between the early onset of pathology and initial diagnosis since composition and structural cartilage changes can be detected (via MRI) alongside potential microscopic bone remodelling and osteophytosis (via CT) [13, 94–96]. However, given the scope of the study, MRI or CT imaging was not considered feasible.

## Future work

It is anticipated that GenPup-M will be a free, open-access owner-reported mobility questionnaire that is easy to use. It will be dog-centred and require no specialised equipment (e.g. pressure/force plates). Until now, there has been no other mobility CMI that has been validated for use on healthy dogs without mobility impairments. Hence, GenPup-M has been created to meet this requirement. GenPup-M, like many other mobility CMIs, can allow the assessment of natural canine behaviours within a home environment, making it useful for future research investigating mobility impairments (such as the Generation Pup™ study). GenPup-M can also be distributed to veterinary practices for completion by owners at yearly vaccinations/health checks and be used throughout the dog's life. Finally, there is scope to use GenPup-M alongside its use for canine fosters within the Dogs Trust and similar rehoming charities.

A collaboration between the University of Liverpool and Dogs Trust is facilitating the development of GenPup-M into an open-access App for owners to use. The authors feel the development of this tool will make a positive contribution to the dog-owning population, and the veterinary profession and improve canine health and welfare. Preliminary data from this study prompted the research team to extend data collection from this existing group of dogs (n = 62) and an "at-risk" breed (n = 60) to examine longitudinal changes in mobility across three years.

## Conclusion

The objective of this study was to validate a novel owner-reported questionnaire (GenPup-M) by using a previously validated veterinary clinical examination sheet [54], the LOAD questionnaire, and quantitative gait analysis in 62 dogs with/without mobility impairments. Results of the study found that GenPup-M positively correlated with all three objective measures of canine mobility and consequently provided criterion and construct validity for GenPup-M. Owner-reported CMIs allow a non-invasive scoring system that veterinary surgeons and owners can use to allow communication and longitudinal assessment of a dog's mobility. Furthermore, previous research has shown that CMIs prove useful screening diagnostic tools, with dogs scoring poorly underdoing further diagnostic imaging. GenPup-M can be used at vaccinations or health checks and it is hoped the development of the smartphone App will facilitate its use and increase owner compliance.

## Supporting information

**S1 Fig. The pre-validated veterinary clinical examination sheet used for this project.** The clinical examination sheet was validated by Harris *et al*. (2018). The tool was used to assess joint function score (JFS) adapted by Impellizeri et al (2000) where '0' represented the normal range of motion (ROM) and '4' signified three to four joints abnormalities and a pain response was elicited upon touching the joint. Mobility/lameness scores in this validated clinical examination (0–10) were adapted from previous studies (Vasseur, 1993) to determine the dog's ROM. Visual gait analysis can be conducted in the consultation room during history taking to reveal subtle lameness, including assessment of stand-up/lie-down phases (Hazewinkel, 2003). (PDF)

**S2 Fig. The GenPup-M questionnaire has been developed as a mobility questionnaire which aims to accurately identify subtle changes relating to canine mobility.** GenPup-M can collect mobility data from dog owners at repeated time points and be used in clinical settings or within the Dogs Trust Generation Pup™ longitudinal study to identify risk factors indicative of early reductions in mobility. Hence, early interventions can be implemented, improving canine health and welfare. (PDF)

**S3 Fig. Raw and normalised values for peak vertical forces (PVF) and absolute and relative speed.** (PDF)

**S4 Fig. Kruskal-Wallis outputs from the comparison of body mass with normalised Peak Vertical Force (PVF) ratio in non-mobility impaired versus mobility impaired cohorts.** (PDF)

**S5 Fig. The scoring system used for GenPup-M to create a total clinical examination score used for correlation of GenPup-M responses to LOAD questionnaire, clinical examination and Peak Vertical Force (PVF).** Highest score for the questionnaire was 108 and the severity was categorised as follows: score of 0–27 = No concerns; score of 28–54 = Mild concerns; score of 55–82 = Moderate concerns; score of 82–108 = Severe concerns. Questions 12–21 were used for the correlation statistics due to these being owner reported scale/rank questions relating to their dog's activity levels, willingness to exercise and general demeanour. (PDF)

## Acknowledgments

The authors are grateful for permission to use the Liverpool Osteoarthritis in Dogs (LOAD) questionnaire, a clinical metrology instrument developed by the University of Liverpool and distributed exclusively by Elanco Animal Health for the evaluation of osteoarthritis and outcomes in this study. The authors would like to thank Hannah Seymour and Hannah Clark who were students at the University of Liverpool and helped to collect and clean the gait data. Thank you to the technical team in the William Henry Duncan Building, and thank you to all the owners who participated their dogs in this study.

## Author Contributions

**Conceptualization:** Karl T. Bates, Lauren K. Harris, Andrew W. Tomlinson, Jane K. Murray, Eithne J. Comerford.

**Data curation:** Natasha L. Clark, Karl T. Bates, Eithne J. Comerford.

**Formal analysis:** Natasha L. Clark, Karl T. Bates, Jane K. Murray.

**Funding acquisition:** Karl T. Bates, Jane K. Murray, Eithne J. Comerford.

**Investigation:** Natasha L. Clark.

**Methodology:** Natasha L. Clark, Karl T. Bates, Eithne J. Comerford.

**Project administration:** Natasha L. Clark.

**Software:** Natasha L. Clark, Karl T. Bates, Lauren K. Harris.

**Supervision:** Karl T. Bates, Lauren K. Harris, Andrew W. Tomlinson, Jane K. Murray, Eithne J. Comerford.

**Validation:** Natasha L. Clark, Eithne J. Comerford.

**Visualization:** Eithne J. Comerford.

**Writing – original draft:** Natasha L. Clark, Karl T. Bates, Eithne J. Comerford.

**Writing – review & editing:** Natasha L. Clark, Karl T. Bates, Lauren K. Harris, Andrew W. Tomlinson, Jane K. Murray, Eithne J. Comerford.

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
