## [Decision Letter · Decision Letter 0]

29 May 2023

PONE-D-23-07825GenPup-M: a novel validated owner-reported clinical metrology instrument for detecting early mobility changes in dogsPLOS ONE

Dear Dr. Clark,

Thank you for submitting your manuscript to PLOS ONE. After careful consideration, we feel that it has merit but does not fully meet PLOS ONE’s publication criteria as it currently stands. Therefore, we invite you to submit a revised version of the manuscript that addresses the points raised during the review process.

We look forward to receiving your revised manuscript.

Kind regards,

Juan J Loor

Academic Editor

PLOS ONE

Reviewers' comments:

Reviewer's Responses to Questions

**Comments to the Author**

1. Is the manuscript technically sound, and do the data support the conclusions?

Reviewer #1: Partly

Reviewer #2: Yes

2. Has the statistical analysis been performed appropriately and rigorously? 

Reviewer #1: Yes

Reviewer #2: Yes

3. Have the authors made all data underlying the findings in their manuscript fully available?

Reviewer #1: No

Reviewer #2: Yes

4. Is the manuscript presented in an intelligible fashion and written in standard English?

Reviewer #1: Yes

Reviewer #2: Yes

5. Review Comments to the Author

Reviewer #1: I appreciate the opportunity to review this manuscript. The authors present an interesting piece of research and are commended for addressing the important topic of early detection of mobility impairment conditions. The study used different approaches to validate a new clinical metrology instrument, the GenPup-M questionnaire.

I have some comments that I believe could improve the paper's overall quality. Please find them below. On a personal note, I prefer to have the line number, as it makes it easier to reference relevant text sections.

Abstract:

Results paragraph, line 1 – PCA is only used once in the abstract. It does not make sense to add an abbreviation.

Results paragraph, lines 9-10: "Analyses of PVF showed that non-mobility impaired dogs unevenly distributed their weight across all thoracic and pelvic compared to mobility impaired dogs" – This sentence is unclear. Does it make sense? Dogs with impaired mobility had a more balanced weight-bearing distribution?

Conclusion and Clinical Relevance paragraph, lines 1-3: "GenPup-M is clinical metrology instrument (CMI) that can be completed by dog owners to detect all mobility impairments, including those that are early in onset." I would take more care with these statements. Based on your work and results, can you conclude that the GenPup can detect all mobility impairments? You don't have a definitive diagnosis for the animals in the sample, which is a major limitation. How can you state that all impairments can be detected? Perhaps all the ones included in the study, but you still don't know what those are. Also, can you state that it could detect impairments at an early age? The mean age of your sample is 6yo, which is not that young. And you do not provide information on the age of the dogs in the non-impaired group.

Conclusion and Clinical Relevance paragraph, lines 8-9: "which veterinary surgeons can use in conjunction owners". The sentence does not make sense.

Introduction:

Page 3, paragraph 1, lines 3-5 – I agree that detecting mobility impairments due to neurological disease is very important. However, this is the only reference in the manuscript to neurological disease, and although a neuro exam was performed, it seems that they were all normal. That being said, is this section relevant?

Page 3, paragraph 2, line 3 – Based on the provided reference, the values presented are for OA alone, not orthopedic conditions as a whole.

Page 3, paragraph 3, line 4 – "diseases (such as OA) identifiable", are identifiable?

Page 4, paragraphs 2-3 – This is too much unrelevant information. The study aims to validate a new CMI. With that in mind, the introduction should focus more on that process and the required steps for doing so (e.g., see 10.1371/journal.pone.0058125.t002, 10.3390/ani12202808, or 10.1186/s13028-017-0311-2). In this context, this amount of information on NRS and VAS scales and kinetic and kinematic evaluations is unnecessary.

Page 5, paragraph 2, lines 6-7 – I suggest you check the reference 10.1186/s12917-020-02647-2.

Page 5, paragraph 3, lines 2-5 – A recent study (10.1371/journal.pone.0280912) reported data on the LOAD and COI in dogs with acute and chronic CCL disease/rupture.

At this point, something has to be said about the GenPup-M questionnaire. Where does it come from? How were the questions defined at the final ones set? Who conducted this process? Why do we need a new CMI when we have several that have been extensively validated and worked on? Why is it more advantageous to use the GenPup-M and not the LOAD? Why did the authors choose to investigate the ability of the GenPup-M to meet the study's objectives? Why not do this with the LOAD?

M&M:

Page 8, clinical examination section, paragraph 1, line 1 – "Each clinical examination sheet Harris et al. [49] (S1 Fig.). contained". This section does not make sense.

Page 8, clinical examination section, paragraph 2, lines 1-2 – How many different examiners? Are they board certified? Was inter-observer repeatability assessed?

Page 10, GenPup-M section, paragraph 2 – references are needed.

Page 11, line 2 – A "." is missing at the end of the sentence.

Results:

Page 11, Table 1 – BCS was assessed and should be reported as an important clinical finding, particularly regarding OA patients. We also need to have this information broken down into groups. For example, one of your aims with the GenPup-M is to aid in the early detection of mobility impairments. But your sample mean age is 6yo, a similar age to 10.1186/s12917-020-02647-2, referring to dogs with radiographic hip OA, so it would not be an early detection. In which group were the younger animals? I would guess they were in the non-impaired group. Also, no actual diagnosis is presented, which is a major limitation. And what were the results of the neuro exam?

Page 13, paragraph 2, lines 2-3 – "Age had a significant positive correlation with an increase in overall clinical examination scores" this finding seems consistent with my previous comment on age.

Page 13, paragraph 3, line 3 – χ2.

Page 14, GenPup-M and LOAD responses section, paragraph 1 – These findings are similar to LOAD (see 10.1371/journal.pone.0058125.t002 and 10.3390/ani12202808).

Discussion:

Page 17, paragraph 2, line 1 – CMI abbreviation has been used before.

Page 17, paragraph 2, lines 11-12 – In this study. How about others?

Page 18, paragraph 2, lines 1-3 – How does this relate to the finding that out of the 31 impaired animals, only 13 (41.9%) had an owner-reported impairment?

Page 18, paragraph 2 and on – Much discussion surrounds kinetic and kinematic analysis. Though interesting, these evaluations are not the focus of the study, particularly since a definitive diagnosis is not presented. A lot more attention should be placed on the validation of Gen-Pup. Also, many references were made to OA, which should be considered a possible diagnosis for many of the animals in the sample. But the diagnosis was not made…

Future work:

Again, why is all of this relevant? Why is it necessary to develop a new CMI? Why should clinicians use the Gen-Pup? Why not use the LOAD, which has been extensively studied? All the data the authors collected to validate the Gen-Pup can be used to further validate the LOAD, even for early detection, if it is the case.

Acknowledgments:

"Liver-pool" twice – it should be Liverpool.

Reviewer #2: The present study has been performed to validate the GEN PUP-M, a novel clinical metrology instrument aimed to detect early mobility impairments in dogs, by comparing results obtained with this new tool with those obtained with a previously validated veterinary clinical examination sheet questionnaire (the LOAD), clinical examination scores and kinetic and kinematic gait analysis.

The study is very well conducted and described, and the obtained results allow us to prospect the availability of a new tool to be used in the clinical practice to detect all canine mobility impairments, including those that are early in onset, thus allowing a timely therapy.

For these reasons, I believe that the paper deserves to be published.

I have only one concern: to my knowledge, the comparison between GEN PUP-M and LOAD is part of the psychometric tests that evaluate criterion and not construct validity. Indeed, criterion validity establishes the validity of a measuring instrument by comparing it with some external criterion (Souza and Silva 2005) (in this case: clinical examination scores, kinetic and kinematic gait analysis, AND LOAD), while construct validity examines whether the instrument detects changes in the construct theoretically conjectured, which provides the strongest evidence for validation (Crellin et al.2007). In the context of the present study, to confirm construct validity, the tool should differentiate dogs with mobility impairments from dogs without mobility impairments.

If the authors agree with this comment, the results of this analysis should be performed and added to the “Construct validity” results session, along with the results of factor analysis, which is actually part of construct validity (indeed, construct validity can also be checked using factor analysis, which distinguishes the underlying dimensions – dimensionality - that establish the relationship between the instrument items). The paragraph “GenPup-M and LOAD responses” (except the sentence on factor analysis) should be moved to the “Criterion validity” results session. When referring to a comparison between GenPup-M and LOAD responses along the manuscript, it should be defined as criterion validity (and not as construct validity).

A couple of other suggestions:

In the abstract, the sentence should be: “Owners were asked to complete THE GENPUP-M ALONG WITH a previously validated mobility questionnaire (Liverpool Osteoarthritis in Dogs (LOAD)) to identify CRITERION validity”.

In the introduction, the Canine Osteoarthritis Staging Tool (COAST) (https://doi.org/10.1016/j.tvjl.2018.02.017)

should be added to the list of the available and validated CMIs.

6. PLOS authors have the option to publish the peer review history of their article (what does this mean?). If published, this will include your full peer review and any attached files.

Reviewer #1: No

Reviewer #2: **Yes: **Giorgia Della Rocca

---

## [Author Response · Author response to Decision Letter 0]

24 Jul 2023

Comments REVIEWER #1:

1. I appreciate the opportunity to review this manuscript. The authors present an interesting piece of research and are commended for addressing the important topic of early detection of mobility impairment conditions. The study used different approaches to validate a new clinical metrology instrument, the GenPup-M questionnaire. I have some comments that I believe could improve the paper's overall quality. Please find them below. On a personal note, I prefer to have the line number, as it makes it easier to reference relevant text sections.

We thank you the Reviewer for their kind and constructive comments. We have implemented line numbers for convenience and have used line numbers to reference the changes within the manuscript throughout this document. 

Abstract

1. Results paragraph, line 1 – PCA is only used once in the abstract. It does not make sense to add an abbreviation. 

Thank you for this suggestion, this had now been removed (lines 51-52)

New sentence: Principal Component Analysis identified GenPup-M had two components with Eigenvalues >1 (“stiffness/ease of movement” and “willingness to be active/exercise”) 

2. Results paragraph lines 9-10: "Analyses of PVF showed that non-mobility impaired dogs unevenly distributed their weight across all thoracic and pelvic compared to mobility impaired dogs" – This sentence is unclear. Does it make sense? Dogs with impaired mobility had a more balanced weight-bearing distribution?

Thank you for highlighting this error within the sentence. The Reviewer is correct, mobility impaired dogs more unevenly distributed their weight compared to non-mobility impaired dogs. This error has been corrected on line 59.

New sentence: Analyses of PVF showed that non-mobility impaired dogs more evenly distributed their weight across all thoracic and pelvic when compared to mobility impaired dogs.

3. Conclusion and Clinical Relevance paragraph, lines 1-3: "GenPup-M is clinical metrology instrument (CMI) that can be completed by dog owners to detect all mobility impairments, including those that are early in onset." I would take more care with these statements. Based on your work and results, can you conclude that the GenPup can detect all mobility impairments? You don't have a definitive diagnosis for the animals in the sample, which is a major limitation. How can you state that all impairments can be detected? Perhaps all the ones included in the study, but you still don't know what those are. Also, can you state that it could detect impairments at an early age? The mean age of your sample is 6yo, which is not that young. And you do not provide information on the age of the dogs in the non-impaired group.

Thank you for raising this concern. We feel that we have now justified the use of GenPup-M for early detection of mobility impairments (e.g. lines 188-196; 215-219; 617-630; 682-701; 739-747). We have also rewritten lines 63-66 within the abstract to reflect the explanations throughout the manuscript. In this study, the authors feel it was not absolutely necessary to have a diagnosis in the mobility impaired cohort, since most dogs presented with mild mobility impairments, thus, a diagnosis may have never been determined in small animal practice for these cases. Moreover, the results of this study demonstrated that GenPup-M positively correlated with a validated clinical examination and quantitative gait analysis, providing some evidence that GenPup-M can identify early abnormalities within a dog’s mobility. Finally, GenPup-M is not intended to be used as a diagnostic tool, but to highlight concerns with a dog’s mobility which owners will then raise with their veterinary surgeon so that further diagnostic tests can be undertake as necessary. We have included a new table (lines 410-416) containing signalment data for both mobility impaired and non-mobility impaired groups, highlighting that mobility impairments were detected in dogs as young as 17 months old. 

4. Conclusion and Clinical Relevance paragraph, lines 8-9: "which veterinary surgeons can use in conjunction owners". The sentence does not make sense.

This sentence has been corrected on lines 70-73. 

New sentence: Owner-reported CMIs such as GenPup-M allow non-invasive scoring systems which veterinary surgeons and owners can use to allow communication and longitudinal assessment of a dogs’ mobility.

Introduction

1. Page 3, paragraph 1, lines 3-5 – I agree that detecting mobility impairments due to neurological disease is very important. However, this is the only reference in the manuscript to neurological disease, and although a neuro exam was performed, it seems that they were all normal. That being said, is this section relevant?

We agree with the Reviewer and have removed this sentence from the introduction. A brief neurological examination was performed on all dogs participating in this study, however, no neurological abnormalities were found to affect mobility or cause an impairment. Thus, since neurological conditions are not the focus of this study, lines 80-82 have been removed. 

2. Page 3, paragraph 2, line 3 – Based on the provided reference, the values presented are for OA alone, not orthopaedic conditions as a whole.

This sentence has been re-worded to focus solely on OA (line 91). 

New sentence: A recent study calculated annual period prevalence at 2.5%, equating to 200,000 dogs in the UK affected by OA, which can cause subsequent mobility impairments. 

3. Page 3, paragraph 3, line 4 – "diseases (such as OA) identifiable", are identifiable?

Thank you for noticing this grammatical error, we have re-worded this sentence on lines 103-104. 

New sentence: Furthermore, it fails to achieve a correlation between identifiable musculoskeletal diseases (such as OA) and clinical joint function. 

4. Page 4, paragraphs 2-3 – This is too much unrelevant information. The study aims to validate a new CMI. With that in mind, the introduction should focus more on that process and the required steps for doing so (e.g., see 10.1371/journal.pone.0058125.t002, 10.3390/ani12202808, or 10.1186/s13028-017 0311-2). In this context, this amount of information on NRS and VAS scales and kinetic and kinematic evaluations is unnecessary.

Thank you for raising this concern about irrelevant information within the introduction. We agree with the Reviewer’s comments and have removed/reduced excessive text relating to NRS, VAS, kinetic and kinematic gait analysis. We have still discussed kinetic and kinematic gait analysis, since this was undertaken during the study and feel it is important for readers to have a basic understanding of kinetic and kinematic gait analysis, and the differences between them. However, these paragraphs have been greatly reduced (lines 122-134). Furthermore, a new paragraph has been included discussing the steps in validating a new veterinary CMI (lines 200-213).

5. Page 5, paragraph 2, lines 6-7 – I suggest you check the reference 10.1186/s12917-020-02647-2.

We thank the Reviewer for making us aware of this study, we have incorporated it into the introduction (lines 169-177) and feel it highlights the importance of GenPup-M as a new CMI. 

6. Page 5, paragraph 3, lines 2-5 – A recent study (10.1371/journal.pone.0280912) reported data on the LOAD and COI in dogs with acute and chronic CCL disease/rupture.

Again, we thank the Reviewer for making us aware of this new study. We feel it has relevance within the Introduction and a new sentence discussing the study has been included (lines 182-187). We have also discussed the limitations of this study and how the use of GenPup-M would differ from LOAD in this scenario. 

7. At this point, something has to be said about the GenPup-M questionnaire. Where does it come from? How were the questions defined at the final ones set? Who conducted this process? Why do we need a new CMI when we have several that have been extensively validated and worked on? Why is it more advantageous to use the GenPup-M and not the LOAD? Why did the authors choose to investigate the ability of the GenPup-M to meet the study's objectives? Why not do this with the LOAD?

Thank you for highlighting the need for a deeper introduction to GenPup-M, its formulation and the contribution we hope that it can make as a new mobility tool. GenPup-M was developed by two of the co-authors (E.C and J.M) with the intention of creating a novel CMI which could be used in a group of dogs from 5 months of age. GenPup-M was initially designed for use in the longitudinal study “Generation Pup” (Murray et al., 2022; https://generationpup.ac.uk/) with the intention of gathering owner-reported information relating to the dog’s mobility at repeated timepoints. Objective validation of GenPup-M has not yet been performed for use in either Generation Pup or the general dog population. Therefore, the aim of this study was to validate the use of GenPup-M (using construct and criterion validity) to provide reassurance that GenPup-M can accurately identify impaired mobility and non-impaired mobility using survey responses of dog owners. GenPup-M questions were based on all current canine mobility CMIs (such as LOAD, CBPI, COI) and questions which were relevant to the early detection of mobility impairments were modified for use in GenPup-M. Furthermore, the authors were interested in the subtle changes associated with early onset mobility impairments (not just OA), and until now, there has been no other mobility CMI that has been validated for use on healthy dogs without mobility impairments. Hence, GenPup-M was created to meet this requirement. We have now added various sentences throughout the manuscript to answer each question as required by the Reviewer (e.g. lines 188-196; 215-219; 617-630; 682-701; 739-747).

Materials and Methods 

1. Page 8, clinical examination section, paragraph 1, line 1 – "Each clinical examination sheet Harris et al. [49] (S1 Fig.). contained". This section does not make sense.

Thank you for highlighting this error within the sentence structure, this has now been amended (line 298)

New sentence: The clinical examination sheet [49] (S1 Fig.) contained a participant identification (PID) number corresponding to the individual dog.

2. Page 8, clinical examination section, paragraph 2, lines 1-2 – How many different examiners? Are they board certified? Was inter-observer repeatability assessed?

The clinical examination was undertaken by N.C who is a veterinary surgeon, but not board certified. However, N.C received initial training by E.C (an ECVS and RCVS specialist in Small Animal Surgery (Orthopaedics)) on how to complete a comprehensive orthopaedic and neurological examination following the validated examination sheet by Harris et al., 2018. Furthermore, E.C was present to provide a second opinion if N.C was concerned. This protocol has been explained within the manuscript on lines 303-304. 

3. Page 10, GenPup-M section, paragraph 2 – references are needed.

The authors were unsure which specific references were required within the text. However, additional references have been included in paragraph 2 (lines 363, 365, 371 and 379). 

4. Page 11, line 2 – A "." is missing at the end of the sentence.

We thank the Reviewer for noticing this missing period mark. We have included it on line 390. 

Results

1. Page 11, Table 1 – BCS was assessed and should be reported as an important clinical finding, particularly regarding OA patients. We also need to have this information broken down into groups. For example, one of your aims with the GenPup-M is to aid in the early detection of mobility impairments. But your sample mean age is 6yo, a similar age to 10.1186/s12917-020-02647-2, referring to dogs with radiographic hip OA, so it would not be an early detection. In which group were the younger animals? I would guess they were in the non-impaired group. Also, no actual diagnosis is presented, which is a major limitation. And what were the results of the neuro exam?

We thank the Reviewer for raising concerns surrounding these deficits of data for the mobility and non-mobility impaired groups within the dataset. We have updated our demographic table to include the two groups (mobility impaired and non-mobility impaired) (lines 410-415), which will hopefully ease readability. BCS and age for each group has been included. We have also provided more information about dogs who presented with definitive diagnosis via veterinary clinical history (n=8) or radiographic evidence (n=6) (lines 406-408). The results of the neurological examination are as follows: “No dogs were found to have neurological deficits resulting in a mobility impairment or reduced quality of life” and this statement appears on lines 420-421. 

2. Page 13, paragraph 2, lines 2-3 – "Age had a significant positive correlation with an increase in overall clinical examination scores" this finding seems consistent with my previous comment on age.

Please see above response for Comment 1 of the Results section. 

3. Page 13, paragraph 3, line 3 – χ2.

Thank you for noticing this error within the text, this has been corrected on line 442. 

4. Page 14, GenPup-M and LOAD responses section, paragraph 1 – These findings are similar to LOAD (see 10.1371/journal.pone.0058125.t002 and 10.3390/ani12202808). 

Yes, we agree with this comment and the findings are discussed on lines 596-600. 

Discussion

1. Page 17, paragraph 2, line 1 – CMI abbreviation has been used before.

Thank you, this abbreviation has been removed from the text (line 578)

2. Page 17, paragraph 2, lines 11-12 – In this study. How about others?

Thank you for raising this point, we have included comparable results from when the LOAD questionnaire underwent another Principle Component Analysis (Walton et al., 2013) (lines 589-594)

New statement: Previous work has shown that component three of the LOAD has identified the “effect of weather” and not “attitude/demeanour” [20]. This difference may be accounted for by a change in participants, since most dogs were only reported to have a mild lameness (n=24), and that the research was completed during spring and summer months, hence, the effect of cold, damp weather was not at the forefront of owner’s interpretations

3. Page 18, paragraph 2, lines 1-3 – How does this relate to the finding that out of the 31 impaired animals, only 13 (41.9%) had an owner-reported impairment?

A discussion has been included on how these results relate to the owner’s previous knowledge of mobility impairments (lines 617-623). We have found that the relatively strong correlations between gait metrics and questionnaire responses suggest that owners can reliably recognise signs of mobility issues in their dog’s habitual movement patterns. Only 5 out of the 13 dogs had moderate or severe mobility impairments with visual gait abnormalities, thus, 8 dogs had owners who were able to identify subtle changes in their dog’s mobility based off questions relating to their dogs’ attitude to exercise and/or demeanour.

4. Page 18, paragraph 2 and on – Much discussion surrounds kinetic and kinematic analysis. Though interesting, these evaluations are not the focus of the study, particularly since a definitive diagnosis is not presented. A lot more attention should be placed on the validation of Gen-Pup. Also, many references were made to OA, which should be considered a possible diagnosis for many of the animals in the sample. But the diagnosis was not made. 

We agree with the Reviewer and have reduced the amount of kinetic and kinematic discussion between lines 632 and 680. We have replaced this with justification on why it may not have been important to have a definitive diagnosis, only presumptive. We also discuss the results of this project in relation to validating GenPup-M for use in small animal practice. 

Future work 

1. Again, why is all of this relevant? Why is it necessary to develop a new CMI? Why should clinicians use the Gen-Pup? Why not use the LOAD, which has been extensively studied? All the data the authors collected to validate the Gen-Pup can be used to further validate the LOAD, even for early detection, if it is the case.

We have explained throughout the manuscript why GenPup-M is different to all other CMIs as its use begins with the early detection and longitudinal monitoring of normal dogs (Gen-Pup) and the onset of canine mobility impairments. We have also included a discussion (lines 737-747) on how GenPup-M is different from current CMIs (e.g. LOAD) and how GenPup-M can be used in dogs without mobility impairments from the age of 5 months old to allow early detection and intervention if a change in mobility arises. Furthermore, GenPup-M can allow the assessment of natural canine behaviours within a home environment and can be completed by owners at yearly vaccinations/health checks and to be used throughout the dog’s life. Finally, the LOAD questionnaire is commonly used in dogs already diagnosed with chronic osteoarthritis, whereas GenPup-M can be used in healthy dogs to monitor signs associated with ageing. 

Acknowledgements 

1. "Liver-pool" twice – it should be Liverpool.

We thank the Reviewer for noticing this grammatical error. Both instances have now been changes (lines 773-776). 

Comments REVIEWER #2:

The present study has been performed to validate the GEN PUP-M, a novel clinical metrology instrument aimed to detect early mobility impairments in dogs, by comparing results obtained with this new tool with those obtained with a previously validated veterinary clinical examination sheet questionnaire (the LOAD), clinical examination scores and kinetic and kinematic gait analysis. The study is very well conducted and described, and the obtained results allow us to prospect the availability of a new tool to be used in the clinical practice to detect all canine mobility impairments, including those that are early in onset, thus allowing a timely therapy. For these reasons, I believe that the paper deserves to be published. 

1. I have only one concern: to my knowledge, the comparison between GEN PUP-M and LOAD is part of the psychometric tests that evaluate criterion and not construct validity. Indeed, criterion validity establishes the validity of a measuring instrument by comparing it with some external criterion (Souza and Silva 2005) (in this case: clinical examination scores, kinetic and kinematic gait analysis, AND LOAD), while construct validity examines whether the instrument detects changes in the construct theoretically conjectured, which provides the strongest evidence for validation (Crellin et al. 2007). In the context of the present study, to confirm construct validity, the tool should differentiate dogs with mobility impairments from dogs without mobility impairments. If the authors agree with this comment, the results of this analysis should be performed and added to the “Construct validity” results session, along with the results of factor analysis, which is actually part of construct validity (indeed, construct validity can also be checked using factor analysis, which distinguishes the underlying dimensions – dimensionality - that establish the relationship between the instrument items). The paragraph “GenPup-M and LOAD responses” (except the sentence on factor analysis) should be moved to the “Criterion validity” results session. When referring to a comparison between GenPup-M and LOAD responses along the manuscript, it should be defined as criterion validity (and not as construct validity). 

We thank the Reviewer for their comments and we have investigated the references provided. There seems to be dissimilarity among veterinary researchers as to what is defined as “construct validity” and what is known as “criterion validity”. Reviewer #2 is correct that the LOAD questionnaire could form part of the criterion validity. However, the authors used similar methodology to Walton et al., 2013 when validating the LOAD for use in chronic OA. In the Walton et al. study, data from force platforms (PVF) and activity monitors were compared to LOAD scores and used to determine criterion validity with the definition being “Criterion validity is a test of how well an instrument correlates with a standard, external measure of the disease”. 

Furthermore, Walton et al. tested construct validity “primarily, by comparing LOAD, CBPI and HCPI scores against each other. Additionally, factor analysis was performed for all CMIs and reported for the first time for LOAD, and was compared against that previously reported for CBPI and HCPI”. Moreover, Walton et al. (2013) (10.1371/journal.pone.0058125) explained “Construct validity was also explored using factor analysis. Factor analysis is similar to principal component analysis and is performed in the same way. The aim of factor analysis in construct validation is not necessarily to reduce the data to simplified components for further analysis, but to determine if extracted components can be reasonably explained by the theoretical construct underlying the instrument”. Their definition for construct validity is as follows “Construct validity is a test of how well the instrument’s authors’ theorized construct matches the true construct of what is being measured. It can be tested by administering the instrument to groups of known and differing clinical status (e.g. those with OA and those without); by comparing the results of the instrument with those of other, similar measures; and by factor analysis”. Similarly, Gonçalves et al. (2016) (https://doi.org/10.1590/0103-8478cr20160123) assessed construct validity for the Dog Mobility Scale and suggested future work via qualitative methods of assessing mobility, such as tracking via GPS technology to determine criterion validity. 

Additionally, Reviewer #2 suggests that “In the context of the present study, to confirm construct validity, the tool should differentiate dogs with mobility impairments from dogs without mobility impairments”. GenPup-M has highlighted the ability to differentiate between non-mobility impaired dogs and mobility impaired dogs (of varying degrees) by positively correlating with the subjective (LOAD questionnaire), semi-objective (validated clinical examination) and objective (qualitative gait analysis) forms of canine gait analysis. Hence, the authors feel that GenPup-M has achieved construct and criterion validity via the current methodology. 

2. In the abstract, the sentence should be: “Owners were asked to complete THE GENPUP-M ALONG WITH a previously validated mobility questionnaire (Liverpool Osteoarthritis in Dogs (LOAD)) to identify CRITERION validity”.

As explained above, the authors feel construct and criterion validity has been assessed correctly on this occasion, however, will take on board the Reviewers’ comments and references for future work. The authors thank the Reviewer for noticing the grammatical error, this has been corrected on line 44. 

3. In the introduction, the Canine Osteoarthritis Staging Tool (COAST) (https://doi.org/10.1016/j.tvjl.2018.02.017) should be added to the list of the available and validated CMIs.

We agree with the reviewer that COAST is an important veterinary CMI and has been included on lines 162-163.

---

## [Decision Letter · Decision Letter 1]

21 Aug 2023

GenPup-M: a novel validated owner-reported clinical metrology instrument for detecting early mobility changes in dogs

PONE-D-23-07825R1

Dear Dr. Clark,

We’re pleased to inform you that your manuscript has been judged scientifically suitable for publication and will be formally accepted for publication once it meets all outstanding technical requirements.

Kind regards,

Juan J Loor

Academic Editor

PLOS ONE

Additional Editor Comments (optional):

Reviewers' comments:

Reviewer's Responses to Questions

**Comments to the Author**

1. If the authors have adequately addressed your comments raised in a previous round of review and you feel that this manuscript is now acceptable for publication, you may indicate that here to bypass the “Comments to the Author” section, enter your conflict of interest statement in the “Confidential to Editor” section, and submit your "Accept" recommendation.

Reviewer #1: All comments have been addressed

Reviewer #2: All comments have been addressed

2. Is the manuscript technically sound, and do the data support the conclusions?

Reviewer #1: Yes

Reviewer #2: Yes

3. Has the statistical analysis been performed appropriately and rigorously? 

Reviewer #1: Yes

Reviewer #2: Yes

4. Have the authors made all data underlying the findings in their manuscript fully available?

Reviewer #1: Yes

Reviewer #2: Yes

5. Is the manuscript presented in an intelligible fashion and written in standard English?

Reviewer #1: Yes

Reviewer #2: Yes

6. Review Comments to the Author

Reviewer #1: Thank you for all the work in reviewing the manuscript.

I am happy to recommend it for publication.

Reviewer #2: Dear Authors,

many thanks for addressing my concerns.

I believe that another good aspect of the review process is the possibility for the reviewer to improve its/her knowledge!

7. PLOS authors have the option to publish the peer review history of their article (what does this mean?). If published, this will include your full peer review and any attached files.

Reviewer #1: No

Reviewer #2: **Yes: **Giorgia della Rocca

---

## [Editor Report · Acceptance letter]

18 Sep 2023

PONE-D-23-07825R1 

GenPup-M: a novel validated owner-reported clinical metrology instrument for detecting early mobility changes in dogs 

Dear Dr. Clark:

I'm pleased to inform you that your manuscript has been deemed suitable for publication in PLOS ONE. Congratulations! Your manuscript is now with our production department. 

Kind regards, 

on behalf of

Dr. Juan J Loor 

Academic Editor

PLOS ONE